# Dreaming the Sound of Contact: Leveraging Video and Audio Generation for Zero-Shot Force-Aware Manipulation

*Abstract*— Recent advances in video generation enable learning robot manipulation trajectories from generated videos. However, these approaches produce purely kinematic trajectories that lack force information, leading to failure in contact-rich tasks where appropriate contact forces are essential for success. Generated audio carries a complementary and underexplored signal: contact sounds encode force dynamics that video alone cannot capture. We present a pipeline that jointly leverages generated video and audio to recover both motion trajectories and contact force profiles from a single task description. We execute these force-aware trajectories on a Franka Panda robot using a closed-loop force regulator that tracks the audio-derived force profile during contact. Real-robot experiments on whiteboard wiping and vegetable peeling demonstrate that our force-aware pipeline enables successful contact-rich manipulation from video generation, where a kinematic-only baseline fails. Project website + Videos: https://dreamingcontact.github.io/.

## I. INTRODUCTION

Recent video generation models [1], [2] have enabled zero-shot robot manipulation, reducing the need for large scale data collection from teleoperation or kinesthetic teaching [3], [4]. However, these approaches usually produce purely kinematic trajectories with no force information, causing failure in contact-rich tasks where appropriate contact forces are essential.

Advanced video generation [5] also produce synchronized **audio**. Since friction forces dissipate mechanical energy by converting it into heat, *sound*, and other forms, louder contact sounds could indicate stronger forces, making audio a natural proxy for contact force detection. We jointly leverage generated video for motion trajectories and generated audio for force profiles, recovering force-aware manipulation trajectories from a single task prompt without physical demonstrations, simulation, or force sensors.

In this work, we develop a method (Fig. 1) that extracts force-aware manipulation trajectories from generated video and audio, specifically leveraging audio loudness as a contact force proxy, and executes them on a real robot with a closed-loop force regulator. We show on three contact-rich tasks: whiteboard wiping, carrot peeling, and lamp button pressing, that our force-aware method achieves 83% success versus 5.6% for the kinematic-only baseline.

## II. RELATED WORK

**Robot Manipulation from Generated Videos.** ATM [6] extracts dense point trajectories from videos for policy learning. Im2Flow2Act [7] uses optical flow for zero-shot cross-embodiment transfer. Dream2Flow [3] and NovaFlow [4]

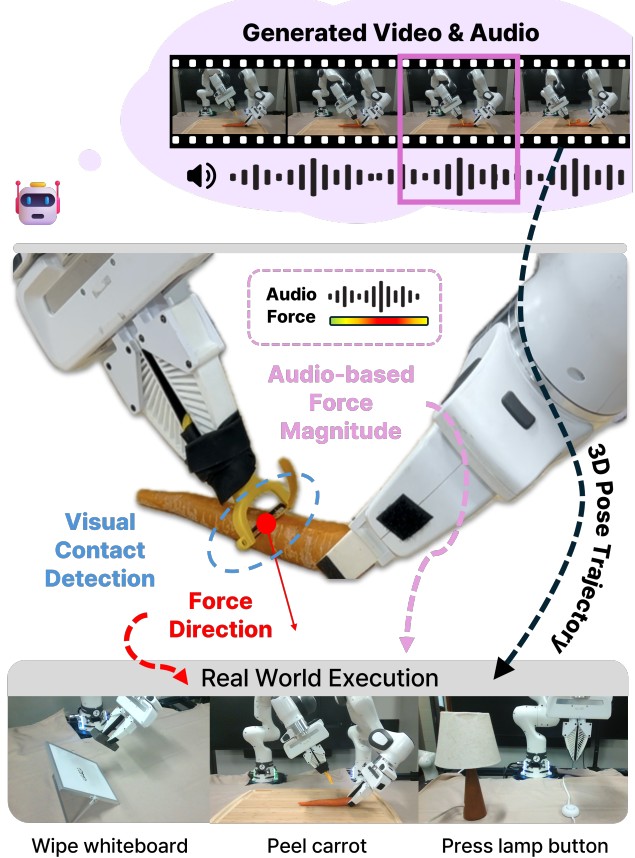

Fig. 1. We leverage generated video and audio as complementary cues to infer both motion and force for contact-rich manipulation.

leverage video diffusion models for zero-shot manipulation. These methods focus exclusively on kinematic information and do not account for contact forces, limiting their applicability to contact-rich tasks.

**Force-Aware Manipulation.** Impedance and compliance control [8] are widely used for contact-rich tasks. ACP [9] learns anisotropic stiffness modulation from demonstrations. Flow with the Force Field [10] learns compliant flow matching policies from force-guided simulation. DexForce [11] extracts force-informed actions from kinesthetic demonstrations with F/T sensors. Our work obtains force from *generated audio* rather than physical demonstrations, simulation, or specialized sensors.

**Audio in Manipulation.** Audio has been explored as a sensing modality for manipulation, including fusing audio with vision and touch [12], learning under occlusion [13], and using contact sounds for dynamic manipulation [14]. These works use real audio captured during execution as

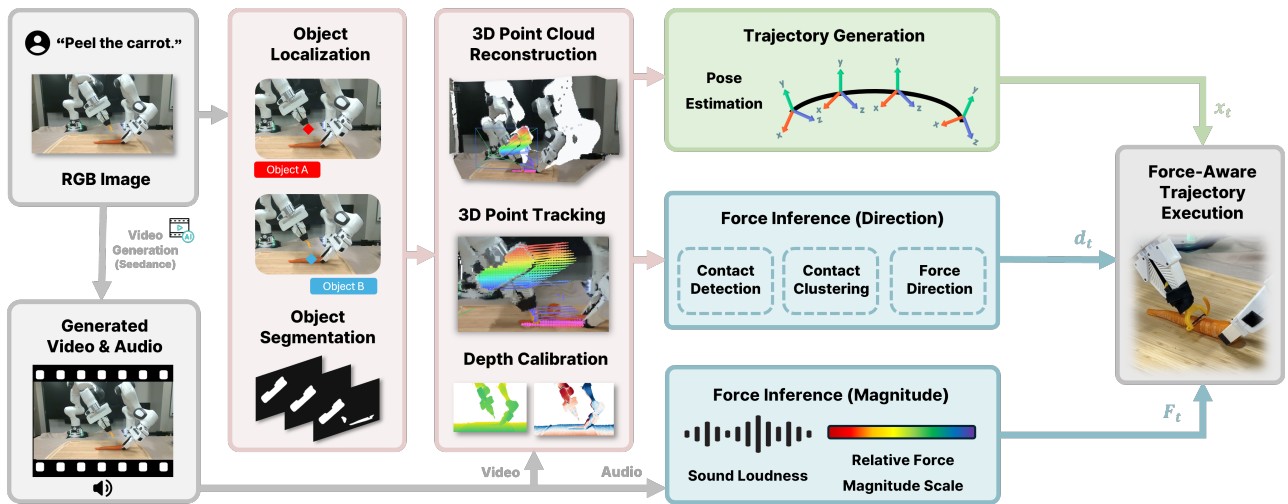

Fig. 2. **Pipeline overview.** Given a generated video with audio, our vision pipeline segments objects (SAM 2), estimates depth (Depth Pro), and tracks 3D points (SpaTracker) to detect contacts and estimate force directions. The audio pipeline extracts loudness as a force magnitude proxy. The combined force-aware trajectory is executed on a Franka Panda with force regulator through a 1 kHz impedance controller.

sensory feedback. In contrast, our audio is *generated* alongside the video, which we use it to extract a force magnitude profile, turning audio into a force specification channel.

## III. APPROACH

### A. Problem Formulation

Our pipeline (Fig. 2) takes as input an initial RGB-D observation $(I_0, D_0)$, a calibrated camera pose ${}^b\mathbf{T}_c$ with respect to the robot base, and a language prompt $p$ describing the task. The exact system and task-specific prompts used for generation are provided in the Appendix. From these inputs, we generate a robot-centric video and recover a force-aware trajectory $\tau = \{(\mathbf{x}_t, \mathbf{d}_t, F_t^*)\}_{t=1}^T$ for execution: end-effector poses $\mathbf{x}_t$, contact directions $\mathbf{d}_t$, and desired contact-force magnitudes $F_t^*$.

### B. Force-Aware Trajectory Extraction from Video and Audio

*1) Video and Audio Generation:* We first synthesize a robot-centric generated video with audio from the initial observation and the task description. Formally, we use a generator $\mathcal{G}$ to produce

$$(V, A) = \mathcal{G}(I_0, p), \qquad V = \{I_t\}_{t=0}^T, \qquad (1)$$

with $I_0$ fixed as the first frame. Since our method is robot-centric, the initial frame must contain the robot end-effector,

$$\Omega_{\text{ee}}(I_0) \neq \emptyset, \qquad (2)$$

with $\Omega_{\text{ee}}(I_0)$ the end-effector region in $I_0$. The generated video $V$ provides the visual trajectory, while the generated audio $A$ provides an aligned signal for force estimation.

*2) Object Segmentation and 3D Reconstruction:* Given the generated video $V$, we first localize both the target object and the gripper in the first frame using MolmoPoint [15], since the interaction occurs between these two entities. We then use the predicted points to initialize SAM 2 [16] for object segmentation over the entire video. Based on the resulting masks, we estimate depth and track 3D points using

TAPIP3D [17]. The predicted depth is calibrated from the first frame using ground-truth depth with a global scaling and offset,

$$z^{\text{cal}}(u, v) = a\, z^{\text{pred}}(u, v) + b, \qquad (3)$$

with fitted calibration parameters $a$ and $b$.

*3) End-Effector Trajectory Estimation:* We next recover the end-effector trajectory before inferring force. Using the gripper mask and TAPIP3D tracks, we collect a set of 3D gripper points in the first frame, $\{\mathbf{p}_i^0\}$, and their correspondences in frame $t$, $\{\mathbf{p}_i^t\}$. Since the visible end-effector can be treated as a rigid object, we estimate the relative rigid transform in the camera frame by

$$(\mathbf{R}_t, \mathbf{t}_t) = \arg \min_{\mathbf{R} \in SO(3),\, \mathbf{t}} \sum_i \left\| \mathbf{p}_i^t - (\mathbf{R}\mathbf{p}_i^0 + \mathbf{t}) \right\|_2^2. \qquad (4)$$

This gives the gripper motion from the first frame to frame $t$ in camera coordinates. Using the calibrated camera pose, we map the trajectory into the robot base frame as

$${}^b\mathbf{T}_t = {}^b\mathbf{T}_c\, {}^c\mathbf{T}_t, \qquad (5)$$

yielding a robot-frame trajectory $\{{}^b\mathbf{T}_t\}_{t=0}^T$ relative to the initial frame.

*4) Audio for Force Extraction:* We estimate force magnitude from the generated audio $A$. We first apply SAM-Audio [18] with a shared system prompt, "The sound of contact between gripper and $\mathcal{O}$," with $\mathcal{O}$ instantiated by the task object. This extracts the contact-related audio while suppressing unrelated background sounds. From the extracted signal, we compute the per-frame loudness in LUFS, denoted by $\ell_t$. We discard low-energy frames with $\ell_t < \ell_{\text{th}}$, treat them as noise, and assign zero force to them. We then convert the remaining loudness values into a relative force magnitude by

$$r_t = 2^{(\ell_t - \ell_0)/10}, \qquad (6)$$

where $\ell_0$ is a fixed reference LUFS level. We then map the valid $r_t$ values into a task-specific force range $[F_{\min}, F_{\max}]$

by min-max normalization,

$$F_t^* = F_{\min} + (F_{\max} - F_{\min}) \frac{\text{clip}(r_t, r_{\min}, r_{\max}) - r_{\min}}{r_{\max} - r_{\min}}, \tag{7}$$

where $r_{\min}$ and $r_{\max}$ are computed from the non-noise portion of the audio. This gives the force magnitude for each frame within the feasible range of the task.

*5) Point Cloud Force Direction:* We estimate force direction only for frames with a nonzero force magnitude inferred from audio. For each such frame, we use the reconstructed gripper and object point clouds to form nearest-neighbor pairs $(\mathbf{q}_i^g, \mathbf{q}_i^o)$ from the gripper to the object, with pairwise vectors

$$\mathbf{v}_i = \mathbf{q}_i^o - \mathbf{q}_i^g. \tag{8}$$

We retain an audio-inferred force only when the gripper and object point clouds are sufficiently close,

$$\min_i \|\mathbf{q}_i^o - \mathbf{q}_i^g\|_2 < d_{\text{th}}. \tag{9}$$

Otherwise, we set the force to zero for that frame. For the remaining frames, we estimate the force direction by averaging the pairwise vectors,

$$\mathbf{d}_t = \frac{\sum_i \mathbf{v}_i}{\|\sum_i \mathbf{v}_i\|_2}. \tag{10}$$

## C. Force-Driven Trajectory Execution

We interpolate the keyframe trajectory to a dense stream of poses $\mathbf{x}_{\text{plan}}(t)$, force directions $\mathbf{d}_t$, and desired magnitudes $F_t^*$, and execute on a Franka Panda using a Cartesian impedance controller with fixed stiffness $k$:

$$\mathbf{F} = k(\mathbf{x}_{\text{cmd}} - \mathbf{x}_{\text{current}}) - \mathbf{D}\dot{\mathbf{x}}. \tag{11}$$

Rather than modulating stiffness, we regulate contact force by driving a virtual-target displacement $\boldsymbol{\delta}$ along the contact direction, so the commanded pose is

$$\mathbf{x}_{\text{cmd}}(t) = \mathbf{x}_{\text{plan}}(t) + \boldsymbol{\delta}, \tag{12}$$

producing contact force $\|F\| \approx k\|\boldsymbol{\delta}\|$ at steady state. We regulate $\boldsymbol{\delta}$ by integrating a normalized force error along the contact direction,

$$\boldsymbol{\delta} \mathrel{+}= K_I \mathbf{d}_t \, \text{clip}\left(\frac{F^* - \hat{F}}{F^*}, -1, 1\right), \tag{13}$$

Here $K_I$ is the integrator gain and $\hat{F}$ is a short-horizon *predicted* force used instead of the instantaneous reading. The clip on the normalized error bounds each update to $\pm K_I$ along $\mathbf{d}_t$, capping the per-tick displacement rate so that force spikes or transient outliers cannot induce a large one-step jump in $\boldsymbol{\delta}$. Sensor filtering and impedance-controller response introduce lag between $\boldsymbol{\delta}$ and the resulting force, so we form

$$\hat{F} = |F| + \dot{F} \cdot \tau, \tag{14}$$

with $|F|$ a low-pass filtered force magnitude and $\tau$ a fixed lookahead, allowing the regulator to anticipate force overshoot and undershoot.

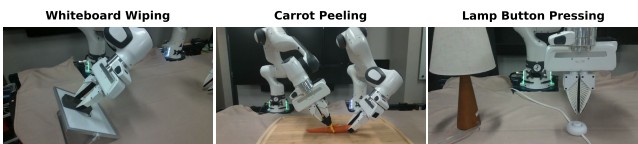

Whiteboard Wiping     Carrot Peeling     Lamp Button Pressing

Fig. 3. Pictures from the three evaluation tasks. **Left**: whiteboard wiping **Center**: carrot peeling **Right**: lamp button pressing

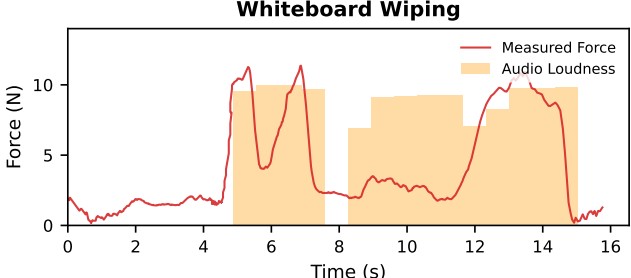

Fig. 4. Whiteboard wiping: audio loudness (orange bars) and measured contact force (red). The regulator drives sustained contact during wiping, with intermittent drops as the eraser transitions between cycles.

## IV. EXPERIMENT

We evaluate three contact-rich tasks on a real Franka Panda robot. For each task, we run six trials with our force-aware pipeline and six with a kinematic-only baseline that uses the same trajectory but zero force specification ($s_f = 0$). The robot runs a Cartesian impedance controller at 1 kHz with fixed stiffness.

### A. Task 1: Whiteboard Wiping

The robot must wipe marker strokes off a whiteboard using an eraser. This requires sustained downward pressure against the board surface while moving laterally.

**Results.** Fig. 4 shows the audio-derived desired force and measured contact force during execution. When the audio signal indicates high contact loudness, the desired force increases to $\sim$10 N, and the regulator drives the virtual target into the surface, resulting in measured forces of 11.4 N. The force-aware execution successfully wipes the board in all six trials. The baseline, lacking a force specification, produces only superficial contact, insufficient to remove marker strokes, and fails all six trials.

### B. Task 2: Carrot Peeling

The robot must peel a carrot using a peeler, requiring sustained contact force against the curved surface while dragging. This task demands both trajectory following and appropriate normal force.

**Results.** Fig. 5 shows the force profiles. The audio signal ramps up as the peeler engages the carrot, with desired force rising gradually from 6 N to $\sim$10 N. The regulator tracks this profile, achieving measured forces up to 10.6 N with a mean of 6.2 N during contact. The force-aware pipeline succeeds in 5 out of 6 trials. The baseline achieves partial contact in some trials due to the trajectory naturally approaching the carrot

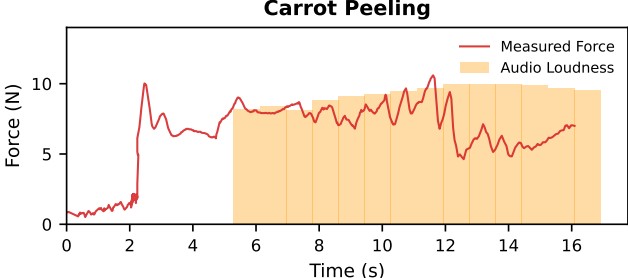

Fig. 5. Carrot peeling: audio loudness (orange bars) stays at a high level as the peeler engages. The measured force (red) closely follows the audio-indicated profile.

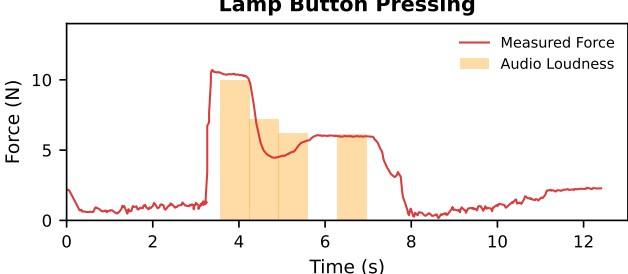

Fig. 6. Lamp button pressing: a sharp audio peak (orange bar) at contact indicates the moment of button engagement. The measured force (red) shows the impulsive contact needed to overcome the spring.

surface, succeeding in 1 out of 6 trials, but with inconsistent peel quality.

### C. Task 3: Lamp Button Pressing

The robot must press a spring-loaded lamp button. The button requires the robot to push through a stiff spring until the mechanism latches. Simply driving downward with maximum stiffness risks triggering the robot's protective reflex due to sudden force spikes at contact.

**Results.** Fig. 6 shows a sharp audio peak at the moment of button contact, producing a brief high-force command. The regulator drives the virtual target downward, achieving measured forces up to 10.7 N, which is sufficient to compress the spring and toggle the lamp. The force profile is concentrated in a short time window ($\sim$3 s), reflecting the impulsive nature of the task. The force-aware pipeline succeeds in 4 out of 6 trials. The baseline, with no additional force applied, fails to push the button past the spring's resistance in all 6 trials.

### D. Quantitative Summary

Table I summarizes the results. The force-aware pipeline achieves a combined success rate of 15/18 (83%), compared to 1/18 (5.6%) for the baseline. The improvement is pronounced across all three tasks. For whiteboard wiping, force regulation provides the sustained contact needed for erasing. For carrot peeling, it reduces sensitivity to motion and depth errors that can cause either weak contact or excessive penetration. For lamp pressing, it supplies the brief, strong force needed to overcome the spring-loaded mechanism. The force tracking RMSE measures how closely the measured

### TABLE I
#### FORCE-AWARE (OURS) VS. KINEMATIC-ONLY BASELINE

|  | Wiping | Peeling | Pressing |
|---|---|---|---|
| Success (Base) | 0/6 | 1/6 | 0/6 |
| Success (Ours) | 6/6 | 5/6 | 4/6 |
| Tracking RMSE (N) | 4.93 | 2.06 | 1.22 |

contact force follows the audio-derived desired profile during active regulation. Carrot peeling and lamp pressing achieve low RMSE (2.06 N and 1.22 N), indicating accurate force tracking. Whiteboard wiping shows higher RMSE (4.93 N) because the eraser intermittently lifts off the board surface between wiping strokes, causing transient force drops despite a sustained desired force—yet the method still achieves 6/6 success. Across tasks with diverse contact dynamics, such as sustained friction (wiping), continuous pressure (peeling), and impulsive contact (pressing), the audio provides a viable force proxy.

## V. DISCUSSION & CONCLUSION

We presented a pipeline for extracting force-aware manipulation trajectories from video and audio generation. Instead of relying solely on vision tracking, we leverage audio as a proxy for recovering force profiles. Experiments demonstrate that force awareness is essential for these tasks and that audio provides a viable proxy for contact force magnitude. Future work includes incorporating real-time audio feedback during execution for closed-loop force adaptation.

## APPENDIX

**System prompt**

```
Video: The camera remains stationary while <TASK
SPECIFIC PROMPT>.
Audio: Emphasize the sound of physical
interaction and contact force central to
the action, while suppressing all ambient,
background, and unrelated sounds.
```

**Task-specific prompt**

*Task 1: Whiteboard Wiping*

```
The gripper, holding the eraser, slowly wipes the
black marks off the whiteboard, then retracts at
the end.
```

*Task 2: Carrot Peeling*

```
The left gripper, holding the yellow peeler,
slowly peels off a layer of the carrot skin,
while the right gripper remains still.
```

*Task 3: Lamp Button Pressing*

```
The gripper performs a single press on the white
button on the table and then rises back up, while
remaining closed.
```

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
