# OpenReview forum: "Dreaming the Sound of Contact: Leveraging Video and Audio Generation for Zero-Shot Force-Aware Manipulation"
_IEEE.org/ICRA/2026/Workshop/Manipulation_Robustness — ICRA 2026_

### Official Review · Reviewer_7uKa · 2026-05-04
**Novel Audio-as-Force-Spec Idea**

**Rating:** 8
**Confidence:** 4

**Review:**

The paper proposes using generated audio (from joint video-audio generators) as a proxy for contact-force magnitude in zero-shot manipulation. A vision branch (SAM 2 + TAPIP3D + Depth Pro) recovers the end-effector trajectory and contact direction; an audio branch (SAM-Audio + LUFS loudness) supplies a desired force profile; a closed-loop regulator on top of a 1 kHz Cartesian impedance controller tracks it via virtual-target displacement. Real-robot results on three tasks (wiping, peeling, button-pressing) report 15/18 success vs. 1/18 for a kinematic-only baseline.

The core idea — turning generated audio into a force specification channel rather than feedback — is novel and timely, and the pipeline is engineered carefully. However, there are several limitations that constrain this method's practice in complex tasks, which can be improved in the future.

Limitations

Eq. (7) min-max normalizes loudness into a task-specific range [Fmin⁡, Fmax⁡] that must be supplied per task. The audio therefore provides only the temporal shape of the force profile; the magnitude scale is a hand-set prior. The paper should state this much more plainly — as written, it reads as if force is recovered end-to-end from audio.

The mapping from loudness to force is heuristic. The exponent in rt=2(ℓt−ℓ0)/10r_t = 2^{(\ell_t - \ell_0)/10}
rt​=2(ℓt​−ℓ0​)/10 is not justified physically, and there is no ablation on the choice. Some sensitivity analysis would be valuable.

Suggestions

Generation reliability is not reported. Video/audio generators frequently produce physically implausible or temporally misaligned outputs. The paper does not say how many (V,A) pairs had to be re-rolled, or what the selection protocol was. This materially affects how the success rates should be interpreted. Please make the impact of artifacts from generated audio be presented clearer.

Questions

How sensitive are results to the choice of [Fmin⁡,Fmax⁡]? Would a single fixed range work across tasks?
What fraction of generated (V,A) pairs were usable, and what was the selection criterion?

Overall

The idea is creative and the execution is sound for a workshop submission. The main asks for revision are: clearly stating that the per-task force range is a required prior, adding at least one baseline that isolates the audio contribution, and reporting generation reliability and missing hyperparameters.

---

### Decision · Program_Chairs · 2026-05-21

Accept